# Dual-Band Unidirectional Reflectionless Propagation in Metamaterial Based on Two Circular-Hole Resonators

**DOI:** 10.3390/ma11122353

**Published:** 2018-11-22

**Authors:** Guofeng Han, Ruiping Bai, Xingri Jin, Yingqiao Zhang, Chengshou An, Youngpak Lee

**Affiliations:** 1Department of Physics, College of Science, Yanbian University, Yanji 133002, China; 2016010376@ybu.edu.cn (G.H.); ruipingbai@163.com (R.B.); 2Quantum Photonic Science Research Center and Department of Physics, Hanyang University, Seoul 133-791, Korea; yplee@hanyang.ac.kr

**Keywords:** metamaterial, unidirectional reflectionlessness, exceptional point

## Abstract

Dual-band unidirectional reflectionless propagation at two exceptional points is investigated in metamaterial, which is composed of only two gold resonators with circular holes, by simply manipulating the angle of incident wave and distance between two resonators. Furthermore, the dual-band unidirectional reflectionless propagation can be realized in the wide ranges of incident angle from $0^{\circ }$ to $50^{\circ }$ and distance from 255 nm to 355 nm between two resonators. In addition, our scheme is insensitive to polarization of incident wave due to the circular-hole structure of the resonators.

## 1. Introduction

In 1998, Bender et al. showed that non-Hermitian Hamiltonian possessed real and positive eigenvalue spectra as long as they preserved parity-time (PT) symmetry [1,2]. Their researches have attracted numerous attentions on the characteristics of non-Hermitian systems in recent years, especially the existence of exceptional point (EP) [3,4,5,6]. Based on EP, various novel phenomena were founded in a family of artificial structures, such as nonreciprocal light propagation [7,8], absober [9], laser [10,11,12], optomechamically induced transparency [13], unidirectional cloaking [14], unidirectional reflectionlessness [15,16,17], and so on. In addition, some other interesting phenomena, such as near-zero-index wires [18], transformation optics [19], and nanoplasmonic sensor [20] can also be founded in artificial structure. Among them, unidirectional reflectionlessness has been the rapidly growing interest due to its potential applications to optical filters, sensors, diodes, and so on.

So far, unidirectional reflectionlessnesses have been proposed in some non-Hermitian Hamiltonian systems with unbalanced gain and loss [21,22,23,24,25,26,27,28,29,30,31,32], such as passive (no gain) optical waveguide [21], large optical multilayer structure [22], two-layer slab [23], plasmonic waveguide [24,25,26,27], and metamaterial [28,29,30,31,32]. In 2013, Feng et al. [21] experimentally demonstrated unidirectional reflectionlessness in PT symmetry optical waveguide system, where reflection from one port was distinctly diminished. Recently, unidirectional reflectionless propagations were realized in non-PT symmetry plasmonic waveguide systems based on phase coupling [24,25,26]. Further, Zhang et al. [27] realized dual-band unidirectional reflectionlessness based on near-field coupling between a single resonator and the resonant modes of two resonators in a non-Hermitian plasmonic waveguide system. More than that, unidirectional reflectionlessness also can be achieved in metamaterial system. In 2014, Kang et al. [29] proposed an ultrathin hybridized metamaterial to investigate one-way zero reflection yielding a topologically stable sharp phase dislocation at EPs based on near-field coupling. Later, Bai et al. [30] realized the unidirectional reflectionlessness based on near-field coupling of two stacked asymmetric nanostrips. Gu et al. [31] and Bai et al. [32] realized the polarization-insensitivity and polarization-switching unidirectional reflectionlessnesses in metamaterials based on far-field phase coupling between two resonators, respectively. However, the above studies [29,30,31,32] in metamaterials focused on single-band unidirectional reflectionlessnesses, the dual-band scheme was seldom mentioned.

In this work, we propose a novel scheme to realize dual-band unidirectional reflectionlessness in metamaterial based on phase coupling between two circular-hole resonators. The metamaterial system consists of an upper and lower circular-hole gold resonators with different radii. The reflectivities for +z (−z) direction at 262 THz and 276.24 THz are ∼0.63 (∼0) and ∼0 (∼0.62) at two EPs, respectively, and high absorptions of ∼0.86 and ∼0.90 for +z and −z directions can be obtained with both quality factors of ∼20, respectively.

## 2. Structure

The unit cell of the metamaterial structure in Figure 1a consists of an upper and a lower gold resonators with circular holes. The thicknesses of the two resonators are *h* = 30 nm, and the upper and lower radii are *r* = 86 nm and *R* = 112 nm, respectively. Periods *t* are 600 nm in both x and y directions. Two gold resonators are embedded in photopolymer (thickness 405 nm) with the permittivity of 2.4025. The entire structure is placed on glass substrate (thickness 300 nm) with the permittivity of 2.25. Distance *s* between two gold resonators is variable. The permittivity of gold is complied with Drude model with plasmon frequency $\omega_{pl}=1.366\times 10^{16}$ rad/s and collision frequency $\omega_{c}=12.24\times 10^{13}\;s^{-1}$ [33,34]. Numerical simulation is carried out by using the commercial finite difference time domain softwave package (CST Microwave Studio). The boundary condition is the unit cell in x and y directions, while is open in z direction. Figure 1b simply describes the scattering property of the structure, where the amplitudes of +z and −z directions propagating waves are denoted by $S_{+ij}$ and $S_{-ij}$ (i,j = 1,2), respectively.

## 3. Results and Discussions

The proposed structure (Figure 1) can be investigated by the temporal coupled-mode theory (CMT) [35,36,37]. The evolutions of amplitudes $a_{1}$ (upper gold resonator) and $a_{2}$ (lower gold resonator) can be expressed as
(1)$$\begin{array}{c} \frac{da_{1}}{dt}=\left(-i\omega_{1}-\gamma_{1}-\Gamma_{1}\right)a_{1}+S_{+11}\sqrt{\Gamma_{1}}+S_{-12}\sqrt{\Gamma_{1}}-i\kappa a_{2},\\ \frac{da_{2}}{dt}=\left(-i\omega_{2}-\gamma_{2}-\Gamma_{2}\right)a_{2}+S_{-22}\sqrt{\Gamma_{2}}+S_{+21}\sqrt{\Gamma_{2}}-i\kappa a_{1}, \end{array}$$
where $\omega_{1}$ and $\omega_{2}$ are resonant frequencies of the upper and lower gold resonators, and $\gamma_{1}$ and $\gamma_{2}$ are decay rates of the upper and lower gold resonators due to intrinsic loss, respectively. $\Gamma_{1}$ and $\Gamma_{2}$ are dissipation rates of the upper and lower gold resonators due to energy escaping into outside space, respectively. κ is the coupling coefficient between two gold resonators. With the conservation of energy, the amplitudes $S_{+ij}$ and $S_{-ij}$ (i,j = 1, 2) satisfy the following relations
(2)$$\begin{array}{cc} S_{+22}=S_{-22}-\sqrt{\Gamma_{2}}a_{2}, & \;S_{-11}=S_{+11}-\sqrt{\Gamma_{1}}a_{1},\\ S_{+22}=\sqrt{\Gamma_{2}}a_{2}, & \;S_{-11}=\sqrt{\Gamma_{1}}a_{1},\\ S_{+12}=\sqrt{\Gamma_{1}}a_{1}, & \;S_{-21}=\sqrt{\Gamma_{2}}a_{2}. \end{array}$$

In addition, the propagation waves should satisfy the formulas $S_{+21}=S_{+12}e^{i\varphi }$ and $S_{-12}=S_{-21}e^{i\varphi }$ with phase difference φ=ωRe(neff)s/c, $n_{eff}$ and *c* the effective refractive index of incident wave and light velocity in vacuum, respectively. Hence, we can obtain the complex coefficients for +z (−z) direction transmission $t_{+z}$ ($t_{-z}$) and reflection $r_{+z}$ ($r_{-z}$), respectively, as
(3)$$\begin{array}{c} \;t=t_{+z}=t_{-z}=\frac{S_{+22}}{S_{+11}}=\frac{S_{-11}}{S_{-22}}=\frac{\sqrt{\Gamma_{1}}\left(-i\kappa +e^{i\varphi }\sqrt{P}\right)\sqrt{\Gamma_{2}}}{{\left(\kappa +ie^{i\varphi }\sqrt{P}\right)}^{2}+\left[\gamma_{1}+\Gamma_{1}-iM\right]\left[\gamma_{2}+\Gamma_{2}-iN\right]},\\ \;r_{+z}=\frac{S_{-11}}{S_{+11}}=1-\frac{\Gamma_{1}\left[\gamma_{2}+\Gamma_{2}-iN\right]}{{\left(\kappa +ie^{i\varphi }\sqrt{P}\right)}^{2}+\left[\gamma_{1}+\Gamma_{1}-iM\right]\left[\gamma_{2}+\Gamma_{2}-iN\right]},\\ \;r_{-z}=\frac{S_{+22}}{S_{-22}}=1-\frac{\Gamma_{2}\left[\gamma_{1}+\Gamma_{1}-iM\right]}{{\left(\kappa +ie^{i\varphi }\sqrt{P}\right)}^{2}+\left[\gamma_{1}+\Gamma_{1}-iM\right]\left[\gamma_{2}+\Gamma_{2}-iN\right]},\\ \;M=\omega -\omega_{1},N=\omega -\omega_{2},P=\Gamma_{1}\Gamma_{2}. \end{array}$$

So, transmissivity, +z and −z direction reflectivities are $T={\left|t\right|}^{2}$, $R_{+z}={\left|r_{+z}\right|}^{2}$ and $R_{-z}={\left|r_{-z}\right|}^{2}$, respectively. Accordingly, absorptivities for +z and −z directions can be expressed as $A_{+z}=1-T-R_{+z}$ and $A_{-z}=1-T-R_{-z}$, respectively.

Figure 2 shows the reflection, absorption and transmission spectra for +z and −z directions, respectively. Figure 2a is the reflection spectra for +z and −z directions based on numerical simulation (solid line) and analytical calculation (dotted line), respectively. Obviously, the results obtained by analytical calculation are in good agreements with that by numerical simulation. According to numerical simulation, reflectivities in +z (−z) direction are ∼0.63 (∼0) and ∼0 (∼0.62) at 262 THz and 276.24 THz, respectively. Therefore, dual-band unidirectional reflectionlessness is realized in metamaterial based on two gold resonators with circular holes. Moreover, high absorptions of ∼0.86 and ∼0.90 in +z and −z directions can be obtained, respectively, at 276.24 THz and 262 THz, where the transmissions are very low, as shown in Figure 2b. Additionally, the corresponding quality factors of ∼20 can be obtained by formula f/Δf with *f* and Δf the resonant frequency and full width half maximum, respectively. The similar approach has been employed in Reference [38], in which the planar array of resonant metallic cross-shape structure has been used and the absorption properties have been discussed. In our unit-cell, two-circular-hole resonators structure is chosen in view of the insensitivity to polarization and high efficiency on realizing dual-band absorption in two directions. Obviously, absorptions in Reference [38] and our scheme are both insensitive to polarization of the incident wave. A comparison of the work in Reference [38] and ours shows that absorption in one direction is obtained in metallic cross-shape structure and two-direction absorption is obtained in two-circular-hole resonators structure.

To clearly demonstrate the dual-band unidirectional reflectionlessness, we investigate the z-component distributions of electric field of two gold resonators with circular holes for +z (−z) direction at two EPs, as shown in Figure 3. At 262 THz, two gold resonators are excited simultaneously in +z (−z) direction and the induced currents are in the opposite (same) directions, as shown in Figure 3a–d, which means that the phase differences between two gold resonators are ∼π (Figure 3a,b and ∼2π (Figure 3c,d), respectively. Hence, reflections at 262 THz are a high value (black solid line) for +z direction and near zero (red solid line) for −z direction, respectively, based on the Fabry-Pérot resonance coupling between two gold resonators, shown in Figure 2a. At 276.24 THz, two gold resonators are excited simultaneously by incident wave of +z (−z) direction, and the induced currents of two gold resonators are in the same (opposite) directions, shown in Figure 3e–h). This is to say, the phase differences between two gold resonators in +z and −z directions are ∼2π and ∼π, respectively, resulting in near-zero reflection (black solid line) for +z direction and high reflection (red solid line) for −z direction, shown in Figure 2a. Obviously, dual-band unidirectional reflectionless phenomenon appears at two EPs (262 THz and 276.24 THz) when distance *s* = 275 nm.

We also plotted the z-component distributions of magnetic field of the two gold resonators for +z (−z) direction at two EPs, as shown in Figure 4. It is obvious to see that the induced magnetic fields of the two gold resonators are opposite (same) at 262 THz in +z (−z) direction, as shown in Figure 4a–d and the induced magnetic fields are same (opposite) at 276.24 THz in +z (−z) direction, as shown in Figure 4e–h. Obviously, dual-band unidirectional reflectionless phenomenon appears at two EPs based on Fabry-Pérot resonance coupling between two gold resonators. As a result, high two-band absorption can be obtained in view of the low transmission at two EPs, as shown in Figure 2b.

Next, based on scattering matrix [21] *S*
$=\left(\begin{array}{cc} t & r_{-z}\\ r_{+z} & t \end{array}\right),$ we continue to discuss the relevant physics phenomena at two EPs. According to scattering matrix *S*, we obtain the eigenvalues of the scattering matrix *S* as $\lambda_{\pm }=t\pm \sqrt{r_{+z}r_{-z}}.$ Here, when $\sqrt{r_{+z}r_{-z}}=0$, two eigenvalues $\lambda_{\pm }$ coalesce and EP appears. In other words, when $r_{+z}$ or $r_{-z}$ is 0, two eigenvalues coalesce and unidirectional reflectionlessness appears.

Real and imaginary parts of the eigenvalues $\lambda_{\pm }$ as the functions of frequency ω are depicted in Figure 5. From Figure 5a,b, the real parts of the two eigenvalues $\lambda_{\pm }$ coalesce, while the imaginary parts cross at 262 THz and 276.24 THz, respectively. Clearly, the real and imaginary parts of eigenvalues are nonzero at 262 THz and 276.24 THz. In this case, *t* is complex and $\sqrt{r_{+z}r_{-z}}$ is 0. Therefore, the dual-band unidirectional reflectionlessness can be obtained at 262 THz and 276.24 THz.

Furthermore, the phase shift $\phi_{1(2)}$ for the upper (lower) gold resonator is given as [34]
(4)ϕ1(2)=arctanIm(Ts,211(2)/Ts,221(2))Re(Ts,211(2)/Ts,221(2))=(ω−ω1(2))(γ1(2)+Γ1(2)).

The phase difference $\varphi_{all}$ between two gold resonators is composed of the phase shifts of the upper gold resonator, lower gold resonator and between two gold resonators. So, the phase differences $\varphi_{all}$ are $\phi_{1}$ − $\phi_{2}$ + 2ϕ in +z direction and $\phi_{2}$ − $\phi_{1}$ + 2ϕ in −z direction. In our structure, the phase shifts of the upper and lower gold resonators for +z (−z) direction are $\phi_{1}$ = 0 ($\phi_{1}$ = $\frac{\left(\omega_{2}-\omega_{1}\right)}{\left(\gamma_{2}+\Gamma_{2}\right)}$) and $\phi_{2}$ = $\frac{\left(\omega_{1}-\omega_{2}\right)}{\left(\gamma_{1}+\Gamma_{1}\right)}$ ($\phi_{2}$ = 0). Accordingly, the phase differences $\varphi_{all}$ are $\phi_{2}$ + 2ϕ and $\phi_{1}$ + 2ϕ in +z and −z directions, respectively. Thus, by adjusting the frequency of the resonator and distance between two resonators appropriately, we can obtain that $\varphi_{all}$ is ∼π (∼2π) at 262 THz and ∼2π (∼π) at 276.24 THz in +z (−z) direction, respectively, resulting in high (near-zero) reflection and near-zero (high) reflection shown in Figure 2(a). Hence, dual-band unidirectional reflectionlessness at two EPs is realized.

Then, we investigate the influences of incident angle and frequency on reflections (absorptions) in +z and −z directions based on numerical simulation. Figure 6a–d shows reflections (absorptions) in +z and −z directions versus the incident angle and frequency when *s* = 275 nm, respectively. Clearly, low reflection (high absorption) region occur blue-shifts with increasing the incident angle in +z and −z directions, respectively. Moreover, low reflection (high absorption) region in +z direction shown in Figure 6a,c corresponds to high reflection (low absorption) region in −z direction shown in Figure 6b,d around 277 THz. While low reflection (high absorption) region in −z direction shown in Figure 6b,d corresponds to high reflection (low absorption) region in +z direction shown in Figure 6a,c around 260 THz. Obviously, dual-band unidirectional reflectionlessness and dual-band absorption can be obtained in a wide range of incident angle from $0^{\circ }$
to
$50^{\circ }$.

Next, we further study the influence of different distance *s* between two gold resonators on reflections (absorptions) in +z and −z directions. Figure 7 shows the reflections and absorptions as the functions of distance *s* and frequency ω in +z and −z directions when the incident angle is zero. From Figure 7a,c, low reflection and high absorption peaks occur red-shifts with increasing the distance s from 255 nm to ∼315 nm and do not occur shift when the distance *s* over ∼315 nm. While low reflection and high absorption peaks do not occur shift with increasing the distance *s* from 255 nm to ∼295 nm and occur red-shifts with increasing the distance *s* from ∼295 nm to 355 nm, as shown in Figure 7b,d. Moreover, low reflection (high absorption) region for distance *s* from 255 nm to 355 nm in +z direction shown in Figure 7a,c corresponds to high reflection (low absorption) region in −z direction shown in Figure 7b,d around 280 THz. Also, low reflection (high absorption) region for distance *s* from 255 nm to 355 nm in −z direction shown in Figure 7b,d corresponds to high reflection (low absorption) region in +z direction shown in Figure 7a,b around 260 THz. Obviously, dual-band unidirectional reflectionlessness and dual-band absorption can be realized in a wide distance range of 255 nm∼355 nm.

## 4. Conclusions

In summary, we have demonstrated the dual-band unidirectional reflectionless propagation at two EPs in metamaterial by using only two gold resonators with circular holes. Reflectivities for +z and −z directions are ∼0.63 (∼0) and ∼0 (∼0.62) at 262 THz (276.24 THz). Moreover, the dual-band unidirectional reflectionless propagation can be realized in the wide ranges of incident angle from $0^{\circ }$ to $50^{\circ }$ and distance between two gold resonators from 255 nm to 355 nm. Through using the circular-hole structure of the resonators, the scheme is insensitive to polarization of incident wave. Our structure is quite suitable for experimental fabrication using standard nano-fabrication procedures [39]. These results will provide a good platform to realize the extraordinary properties of metamaterial systems with potential applications in the integrated nanophotonic devices, such as optical filters, sensors, and diodes. Especially, it can also be applied to refractive index measurements. In our structure, refractive index of surface medium will impact on the resonance frequency of the structure and the variation of resonance frequency will cause the displacement of the position of unidirectional reflecionlessness. Thus, the change of refractive index can be measured by analyzing the variation of spectrum. In the future, we will study the methods to raise the quality factor of absorption, such as by adopting the high-Q resonator.

## Figures and Tables

**Figure 1 materials-11-02353-f001:**
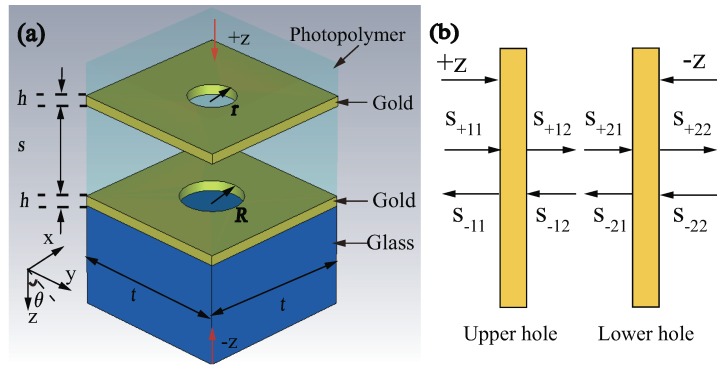
(Color online) (**a**) Schematic diagram of unit cell of the stacked metamaterial structure. The geometric parameters of the structure are *h* = 30 nm, *r* = 86 nm, *R* = 112 nm, *t* = 600 nm. Incident angle θ and distance *s* are variable, respectively. (**b**) Simple schematic diagram of scattering property of the structure. Here, the incident wave is in x–z plane and has a incident angle θ with +z axis, and the electric field is along y axis.

**Figure 2 materials-11-02353-f002:**
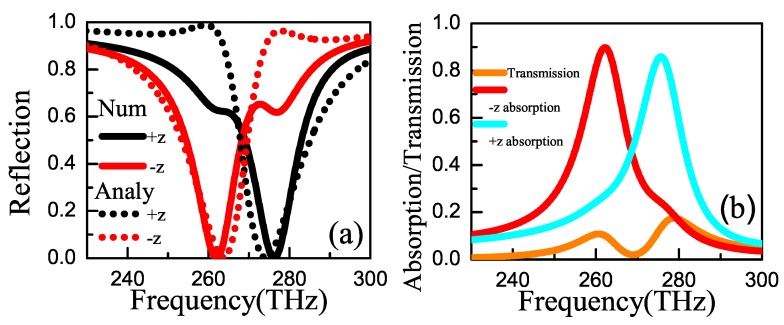
(Color online) (**a**) Numerical simulation (Num) and analytical calculation (Analy) on reflection spectra as the function of frequency ω when *s* = 275 nm and incident angle $\theta =0^{\circ }$. (**b**) Absorption and transmission spectra as the functions of frequency ω. The parameters are $\gamma_{1}=3.209$ THz, $\gamma_{2}=2.150$ THz, $\Gamma_{1}=8.503$ THz, $\Gamma_{2}=8.803$ THz, φ=0.967π and κ=1.765 THz.

**Figure 3 materials-11-02353-f003:**
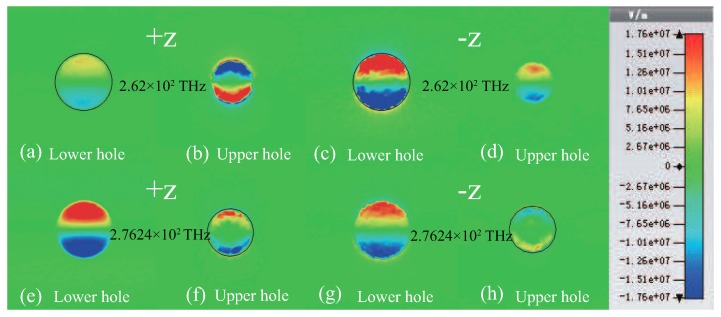
(Color online) z-component distributions of electric field of the two gold resonators at 262 THz (**a**–**d**) and 276.24 THz (**e**–**h**) when distance *s* = 275 nm in +z and −z directions, respectively.

**Figure 4 materials-11-02353-f004:**
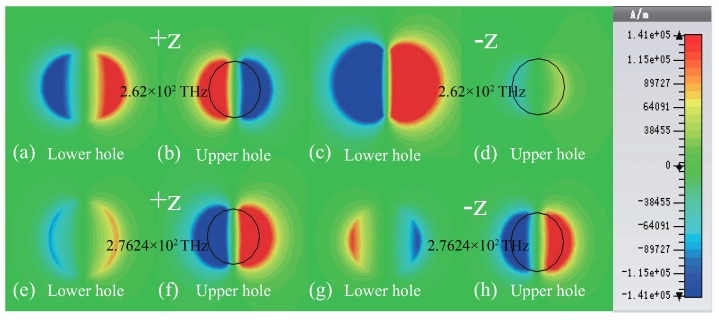
(Color online) z-component distributions of magnetic field of the two gold resonators at 262 THz (**a**–**d**) and 276.24 THz (**e**–**h**) when distance *s* = 275 nm in +z and −z directions, respectively.

**Figure 5 materials-11-02353-f005:**
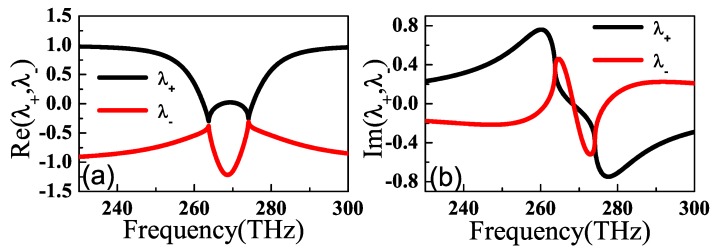
(Color online) Real (**a**) and imaginary (**b**) parts of eigenvalues $\lambda_{\pm }$ of the scattering matrix *S* as the functions of frequency ω when φ=0.967π.

**Figure 6 materials-11-02353-f006:**
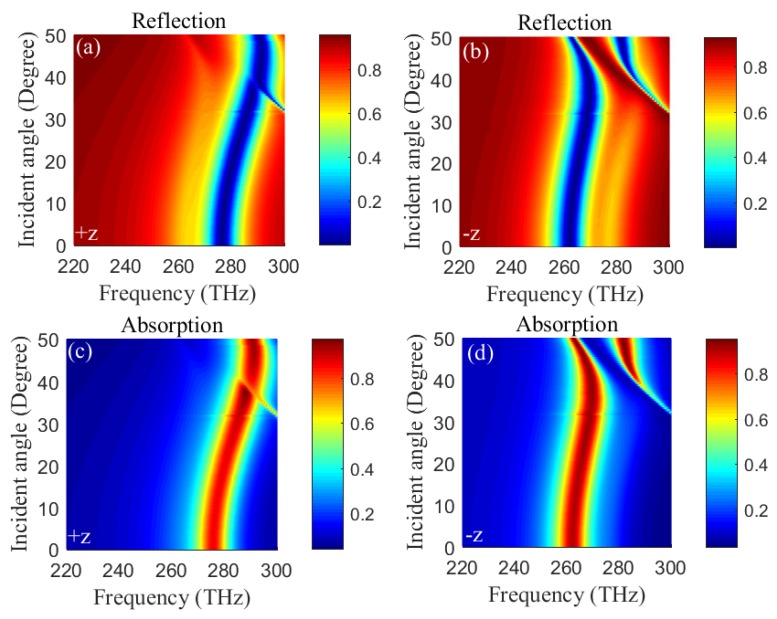
(Color online) Reflections (Absorptions) as the functions of frequency ω and incident angle θ in +z (**a**) and −z (**b**)(+z (**c**) and −z (**d**)) directions when *s* = 275 nm.

**Figure 7 materials-11-02353-f007:**
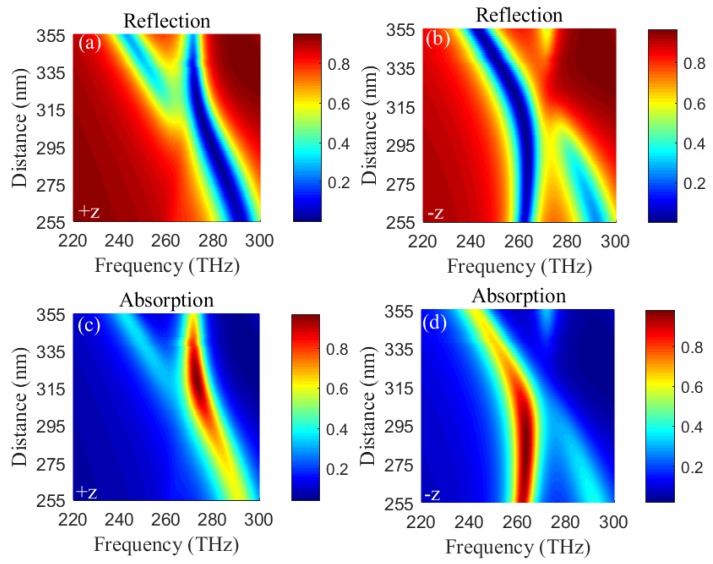
(Color online) Dependences of reflections (absorptions) on distance *s* and frequency ω in +z (**a**) and −z (**b**) (+z (**c**) and −z (**d**)) directions.

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
