# Peer review of "Dual-Band Unidirectional Reflectionless Propagation in Metamaterial Based on Two Circular-Hole Resonators"

_materials, 2018, doi:10.3390/ma11122353_

Round 1
Reviewer 1 Report
Authors propose a novel scheme to realize dual-band unidirectional reflectionlessness in 38 metamaterial based on phase coupling between two circular-hole resonators. The work is interesting, however, I would suggest addition of the supplementary plots into the manuscript. Namely, it would be interesting to have a look at the absorption enhancement of the antennas in consideration
Author Response
Thank you for giving us the chance to respond to the reviewers' comments. We have replied to the comments and revised the manuscript according to these comments.
Thank you for the precious comment for improving the manuscript. We have revised the manuscript according to your comment, as
Point 1: Authors propose a novel scheme to realize dual-band unidirectional reflectionlessness in 38 metamaterial based on phase coupling between two circular-hole resonators. The work is interesting, however, I would suggest addition of the supplementary plots into the manuscript. Namely, it would be interesting to have a look at the absorption enhancement of the antennas in consideration
Response 1: We have added the figures of absorption as the functions of incident angle (distance between two gold hole resonators) and frequency in Fig. 6 (Fig. 7), and inserted the corresponding illustrations, as “Then, we investigate the influences of incident angle and frequency on reflections (absorptions) in +z and –z directions based on numerical simulation. Figs. 6(a) (6(c)) and 6(b) (6(d)) shows reflections (absorptions) in +z and -z directions versus the incident angle and frequency when s=275 nm, respectively. Clearly, low reflection (high absorption) region occur blue-shifts with increasing the incident angle in +z and -z directions, respectively. Moreover, low reflection (high absorption) region in +z direction shown in Fig. 6(a) (6(c)) corresponds to high reflection (low absorption) region in -z direction shown in Fig. 6(b) (6(d)) around 277 THz. While low reflection (high absorption) region in -z direction shown in Fig. 6(b) (6(d)) corresponds to high reflection (low absorption) region in +z direction shown in Fig. 6(a) (6(c)) around 260 THz. Obviously, dual-band unidirectional reflectionlessness and dual-band absorption can be obtained in a wide range of incident angle from 0◦ to 50◦.
Next, we further study the influence of different distance s between two gold resonators on reflections (absorptions) in +z and –z directions. Fig. 7 shows the reflections and absorptions as the functions of distance s and frequency ω in +z and -z directions when the incident angle is zero. From Figs. 7(a) and 7(c), low reflection and high absorption peaks occur red-shifts with increasing the distance s from 255 nm to ∼315 nm and do not occur shift when the distance s over ∼315 nm. While low reflection and high absorption peaks do not occur shift with increasing the distance s from 255 nm to ∼295 nm and occur red-shifts with increasing the distance s from ∼295 nm to 355 nm, as shown in Figs. 7(b) and 7(d). Moreover, low reflection (high absorption) region for distance s from 255 nm to 355 nm in +z direction shown in Fig. 7(a) (7(c)) corresponds to high reflection (low absorption) region in -z direction shown in Fig. 7(b) (7(d)) around 280 THz. Also, low reflection (high absorption) region for distance s from 255 nm to 355 nm in -z direction shown in Fig. 7(b) (7(d)) corresponds to high reflection (low absorption) region in +z direction shown in Fig. 7(a) (7(b)) around 260 THz. Obviously, dual-band unidirectional reflectionlessness and dual-band absorption can be realized in a wide distance range of 255 nm ∼355 nm.” in lines 127-150.
Thank you and please review it again at your earliest convenience.
Sincerely,
Xing Ri Jin
Associate Professor
Department of Physics, College of Science, Yanbian University, Yanji, Jilin 133002, China.
Tel: 86-13704481978
Email:xrjin@ybu.edu.cn, 58387661@qq.com
Reviewer 2 Report
The paper is well written. On the other hand, some modifications are needed as follows:
Section "0. Introduction"
The introduction is clear and all the objectives well stated. Some recent technology developments are missing such as:
_ near-zero-index materials [Near-zero-index wires, Optics express 25 (20), 23699-23708, 2017]
_ graphene [Transformation optics using graphene, Science 332 (6035), 1291-1294, 2011]
_ nanoparticles [Nanoplasmonic sensor for chemical measurements, Optical Sensors 2013 8774, 877411, 2013]
It would be beneficial for the reader if authors include such technologies in the introduction section to have a complete pictures of the state-of-art.
Section "1. Structure"
1) The approach used is similar to [Metamaterial-based wideband electromagnetic wave absorber, Optics express 24 (6), 5763-5772, 2016].
Refer to it and compare your work with it. In particular:
1.1) Explain in details what are the advantages/disadvantages and similarities/differences of your method compared to the above-mentioned.
1.2) A comparison in terms of absorption obtained from the analytical model above mentioned and your model, it would be beneficial for the reader and a good proof-of-concept to confirm the reliability of your approach.
2) A deeper study of the unit-cell is needed.
2.1) Authors should write some lines in this paragraph to explain why they chose to use the proposed structure.
2.2) Explore the capability of the shape and compare yours with the ones contained in the above-mentioned work.
What are the advantages/disadvantages?
Section "2.Results and Discussion"
1) To explore the device behavior, authors can consider the following interesting electromagnetic phenomena:
1.1) electric/magnetic currents [Hybrid bilayer plasmonic metasurface efficiently manipulates visible light, Science Advances, 2(1), 2016]
2.1) surface waves [New absorbing boundary conditions and analytical model for multilayered mushroom-type metamaterials: applications to wideband absorbers, IEEE Trans. Antenn. Propag., 60(12), 5727–5742, 2012]
Include such phenomena in your model and explain how they can affect the device absorption response.
2) The paper lack in application examples. Take into consideration the following: sensing and diagnostics, telecommunications, refractive index measurements, nanoelectronics, antennas and automotive.
I would suggest to create a small paragraph by considering such applications and explaining how you can use your device for them. Please highlight what's new in yours.
Section "3. Conclusions"
1) No limitations of the proposed method have been highlighted.
2) No future improvements/works have been discussed.
Author Response
Thank you for giving us the chance to respond to the reviewers' comments. We have replied to the comments and revised the manuscript according to these comments. Thank you for the precious comments for improving the manuscript. We have revised the manuscript according to your comments, as
The paper is well written. On the other hand, some modifications are needed as follows:
Point 0: Section "0. Introduction"
The introduction is clear and all the objectives well stated. Some recent technology developments are missing such as:
_near-zero-index materials [Near-zero-index wires, Optics express 25 (20), 23699-23708, 2017]
_ graphene [Transformation optics using graphene, Science 332 (6035), 1291-1294, 2011]
_ nanoparticles [Nanoplasmonic sensor for chemical measurements, Optical Sensors 2013 8774, 877411, 2013]
It would be beneficial for the reader if authors include such technologies in the introduction section to have a complete pictures of the state-of-art.
Response 0: We have added three papers [Optics express 25 (20), 23699, 2017], [Science 332 (6035), 1291, 2011] and [Optical Sensors 8774, 877411, 2013] as Refs.[18], [19] and [20] in lines 16~18, as “In addition, some other insteresting phenomena, such as near-zero-index wires[18], transformation optics[19], and nanoplasmonic sensor[20] can also be founded in artificial structure.”
Point 1: Section "1. Structure"
1) The approach used is similar to [Metamaterial-based wideband electromagnetic wave absorber, Optics express 24 (6), 5763-5772, 2016].
Refer to it and compare your work with it. In particular:
1.1) Explain in details what are the advantages/disadvantages and similarities/differences of your method compared to the above-mentioned.
1.2) A comparison in terms of absorption obtained from the analytical model above mentioned and your model, it would be beneficial for the reader and a good proof-of-concept to confirm the reliability of your approach.
2) A deeper study of the unit-cell is needed.
2.1) Authors should write some lines in this paragraph to explain why they chose to use the proposed structure
2.2) Explore the capability of the shape and compare yours with the ones contained in the above-mentioned work.
What are the advantages/disadvantages?
Response 1:
1)-2) We have cited the paper [Optics express 24 (6), 5763, 2016] as Ref. [38] and added some descriptions on their work[38] and ours, as “The similar approach has been employed in Ref.[38], in which the planar array of resonant metallic cross-shape structure has been used and the absorption properties have been discussed. In our unit-cell, two-circular-hole resonators structure is chosen in view of the insensitivity to polarization and high efficiency on realizing dual-band absorption in two directions. Obviously, absorptions in Ref.[38] and our scheme are both insensitive to polarization of the incident wave. A comparasion of the work in Ref.[38] and ours shows that absorption in one direction is obtained in metallic cross-shape structure and two-direction absorption is obtained in two-circular-hole resonators structure. ” in lines 79-85.
Point 2: Section "2.Results and Discussion"
1) To explore the device behavior, authors can consider the following interesting electromagnetic phenomena:
1.1) electric/magnetic currents [Hybrid bilayer plasmonic metasurface efficiently manipulates visible light, Science Advances, 2(1), 2016]
2.1) surface waves [New absorbing boundary conditions and analytical model for multilayered mushroom-type metamaterials: applications to wideband absorbers, IEEE Trans. Antenn. Propag., 60(12), 5727–5742, 2012]
Include such phenomena in your model and explain how they can affect the device absorption response.
2) The paper lack in application examples. Take into consideration the following: sensing and diagnostics, telecommunications, refractive index measurements, nanoelectronics, antennas and automotive.
I would suggest to create a small paragraph by considering such applications and explaining how you can use your device for them. Please highlight what's new in yours.
Response 2 -1):
We have plotted the distributions of magnetic field of the two gold resonators and add some analyses as“We also plotted the z-component distributions of magnetic field of the two gold resonators for +z (-z) direction at two EPs, as shown in Fig. 4. It is obvious to see that the induced magnetic fields of the two gold resonators are opposite (same) at 262THz in +z (-z) direction, as shown in Fig. 4(a) and Fig. 4(b) (Fig. 4(c) and Fig. 4(d)) and the induced magnetic fields are same (opposite) at 276.24THz in +z (-z) direction, as shown in Fig. 4(e) and Fig. 4(f) (Fig. 4(g) and Fig. 4(h)). Obviously, dual-band unidirectional reflectionless phenomenon appears at two EPs based on Fabry-Pérot resonance coupling between two gold resonators. As a result, high two-band absorption can be obtained in view of the low transmission at two EPs, as shown in Fig. 2(b).” in lines 100-107.
Response 2 -2):
We have added some discussions on the application of the relevant research, as“These results will provide a good platform to realize the extraordinary properties of metamaterial systems with potential applications in the integrated nanophotonic devices, such as optical filters, sensors, and diodes. Especially, it can also be applied to refractive index measurements. In our structure, refractive index of surface medium will impact on the resonance frequency of the structure and the variation of resonance frequency will cause the displacement of the position of unidirectional reflecionlessness. Thus, the change of refractive index can be measured by analyzing the variation of spectrum.” in lines 159-165.
Point 3: Section "3. Conclusions"
1) No limitations of the proposed method have been highlighted.
2) No future improvements/works have been discussed.
Response 3:
We have add the expressions, as “In the future, we will study the methods to raise the quality factor of absorption, such as by adopting the high-Q resonator. ” in lines 165-166.
Thank you and please review it again at your earliest convenience.
Sincerely,
Xing Ri Jin
Associate Professor
Department of Physics, College of Science, Yanbian University, Yanji, Jilin 133002, China.
Tel: 86-13704481978
Email:xrjin@ybu.edu.cn, 58387661@qq.com